# Genome-Wide Association Study of *Xian* Rice Grain Shape and Weight in Different Environments

**DOI:** 10.3390/plants12132549

**Published:** 2023-07-04

**Authors:** Nansheng Wang, Wanyang Zhang, Xinchen Wang, Zhenzhen Zheng, Di Bai, Keyang Li, Xueyu Zhao, Jun Xiang, Zhaojie Liang, Yingzhi Qian, Wensheng Wang, Yingyao Shi

**Affiliations:** 1College of Agronomy, Anhui Agricultural University, Hefei 230000, China; 1341489953@stu.ahau.edu.cn (N.W.); 13865512509@163.com (W.Z.); 22721752@stu.ahau.edu.cn (X.W.); zzz4560902@163.com (Z.Z.); 22721703@stu.ahau.edu.cn (D.B.); lky1812966543@163.com (K.L.); 19990114@stu.ahau.edu.cn (X.Z.); xj1259006818@stu.ahau.edu.cn (J.X.); 21721653@stu.ahau.edu.cn (Z.L.); 21720172@stu.ahau.edu.cn (Y.Q.); 2Institute of Crop Sciences, Chinese Academy of Agricultural Sciences, Beijing 100081, China

**Keywords:** rice, grain shape, GWAS, QTLs

## Abstract

Drought is one of the key environmental factors affecting the growth and yield potential of rice. Grain shape, on the other hand, is an important factor determining the appearance, quality, and yield of rice grains. Here, we re-sequenced 275 *Xian* accessions and then conducted a genome-wide association study (GWAS) on six agronomic traits with the 404,411 single nucleotide polymorphisms (SNPs) derived by the best linear unbiased prediction (BLUP) for each trait. Under two years of drought stress (DS) and normal water (NW) treatments, a total of 16 QTLs associated with rice grain shape and grain weight were detected on chromosomes 1, 2, 3, 4, 5, 7, 8, 11, and 12. In addition, these QTLs were analyzed by haplotype analysis and functional annotation, and one clone (*GSN1*) and five new candidate genes were identified in the candidate interval. The findings provide important genetic information for the molecular improvement of grain shape and weight in rice.

## 1. Introduction

Rice (*Oryza sativa* L.) is a globally important food crop, feeding more than half of the world’s population as a staple food [1,2,3]. Hence, rice production is important for global food security, social stability, and economic development. However, rice production is confronted with the challenges of increasing yield loss caused by climate change and gradually decreasing arable land and water resources [4,5]. In addition, the continuous growth of the global population requires further improvement of rice yield in the future [6,7]. In the past decade, the decrease in arable land and the effects of various abiotic stresses have significantly slowed down the average annual growth rate of rice production [8].

Grain shape has important impacts on rice yield and its appearance, processing, cooking, and eating quality, thereby directly affecting the commercial value of rice [9]. In addition, thousand grain weight (TGW), a genetically stable trait, is another important factor affecting rice yield [10]. The factors that determine the TWG include grain length, grain width, and grain thickness [11]. Globally, different countries and regions have different preferences for rice quality traits. For example, people in south and southeast Asia, southern China, the USA, and Latin America prefer long, fine grains with a fluffy and firm texture and medium amylose content. However, people in northern China, Korea, Japan, and parts of the Mediterranean area prefer short, round, soft, and sticky rice grains with a low amylose content [12,13].

Grain shape is an important quantitative agronomic trait in rice, controlled by multiple genes in the canonical form [14,15,16]. Many studies have explored the genetic basis for rice grain shape, resulting in the identification of more than 400 quantitative trait loci (QTLs) in recent decades [13,17,18,19,20]. The common pathways through which these loci regulate grain shape include phytohormone regulation, mitogen-activated protein kinase (MAPK) signaling, transcription factor regulation, G protein signaling, and the ubiquitin-proteasome pathway [21,22]. Instead of acting independently, these regulatory pathways are often intertwined and act in a synergetic manner. For example, *GS5* positively regulates rice grain size by encoding a serine carboxypeptidase [23]. *qTGW3* encodes a kinase similar to the glycogen synthase kinase GSK3, which negatively regulates rice grain size and weight [24]. *OsMAPK6*, *OsMKK4,* and *OsMKP1* are typical genes that regulate grain shape through the MAPK pathway, among which *OsMKK4* regulates rice grain size by mediating cell proliferation [25]. In general, transcriptional regulators are thought to be involved in a variety of regulatory processes. *OsSPL16*/*GW8*, a transcription factor containing the SBP structural domain, regulates rice grain width [26]. *GL7* encodes a homolog of the Arabidopsis LONGIFOLIA protein and regulates the longitudinal elongation of cells [27]. The G protein is an evolutionarily conserved signaling pathway involved in the transmission of extracellular signals to the cell, thereby regulating rice grain shape. The *GS3* gene itself has no effect on grain size, but it can compete with DEP1 or GGC2 for binding Gβ to shorten grain length [28]. Protein ubiquitination is an important regulatory process that affects protein stability, activity, and localization. Among various ubiquitin ligases, *GW2* encodes a cyclic E3 ubiquitin ligase located in the cytoplasm and negatively regulates cell division by anchoring its substrate to the proteasome for degradation [29]. The cloning of these genes and the analysis of their functions have greatly enriched our understanding of the molecular mechanisms underlying the grain shape in rice [30,31]. Since 50% of crop yield loss is caused by abiotic stresses such as drought, salinity, and temperature extremes [32], particularly drought [33], it is important to explore the genetic basis for grain morphology under drought conditions and identify new related QTLs to improve grain yield and quality.

In this study, we tested the grain circumference (GC), grain length (GL), grain length to width ratio (GLWR), grain size (GS), grain width (GW), and thousand grain weight (TGW) of 275 *Xian* rice accessions. To avoid false positives, best linear unbiased predictive values (BLUP) were calculated for these six agronomic traits in different years under drought and normal water treatments by combining phenotypic and genotypic data for a genome-wide association study (GWAS). The GWAS results were further analyzed with various methods. Finally, one cloned gene and five new candidate genes were identified. The findings will help better understand the regulatory mechanisms of rice grain shape, and the new candidate genes may provide an important resource for molecular breeding of rice and the improvement of rice grain shape under drought conditions.

## 2. Results

### 2.1. Distribution and Correlation of Phenotype and Heritability of Grain Shape

The phenotypic traits were significantly different under drought stress (DS) and normal water (NW) in both years (Table 1) (Figure 1a–f). All six traits under DS showed different degrees of reduction compared with those under NW, particularly the TGW. The heritability of GC, GL, GLWR, GS, GW, and TGW was 0.72, 0.94, 0.97, 0.68, 0.95, and 0.71 under DS, and 0.97, 0.98, 0.98, 0.96, 0.98, and 0.99 under NW in different years, respectively (Table 1). A higher heritability indicates high genetic stability, while a lower heritability indicates that the traits are highly influenced by the environment. The variation coefficients of GC, GL, GLWR, GS, GW, and TGW were 8.39–16.44%, 9.84–10.13%, 14.46–16.69%, 10.97–18.08%, and 8.95–9.94%, respectively. Moreover, the wide range of phenotypic variations also enriched the genetic diversity of the studied population. The six-grain shape and weight traits all showed normal distribution, possibly due to the control of these quantitative traits by multiple minor-effect genes (Appendix A).

GL was significantly negatively correlated with GW, with the highest correlation coefficient of −0.433 under NW in 2017, but was significantly positively correlated with other traits. GW was negatively correlated with GLWR and GC but had positive correlations with GS and TGW in both environments, with the highest negative correlation coefficient of −0.835 with GLWR under NW in 2017. GLWR was positively correlated with GS and GC in both environments and only showed a negative correlation to TGW with a coefficient of −0.024 under NW in 2017. GS showed positive correlations with GC and TGW in both environments (Figure 1g–j). These results suggested that there were great phenotypic variations within this population.

### 2.2. Population Structure, Kinship, and LD Decay

The phylogenetic tree (Figure 2a) showed that the studied population had a homogeneous population structure without evident population stratification. The samples were then clustered through principal component analysis (PCA) [34] based on the SNP data using the ‘ggplot2′ package in R software (Figure 2b). The results revealed that the scattered points had a continuous distribution without obvious clustering. In addition, no obvious hotspot was observed on the kinship map (Figure 2c). These results suggested that the tested population had no significant genetic structure or kinship and indicated that the population could meet the requirements of GWAS combined with the distribution of phenotypes.

The LD decay distance determines the minimum number of molecular markers required for association analysis (minimum number of molecular markers = genome size/LD decay distance) and its subsequent resolution. Here, the LD decay distance was approximately 120 kb, as shown in the LD decay plot (Figure 2d), and we obtained a total of 404,411 SNPs, which is perfectly adequate. In general, higher genetic diversity in the population means a shorter LD decay distance, and vice versa [35]. Considering the LD decay distance in rice, adjacent SNPs with spans less than 200 kb [36,37] were defined as one single QTL, and the SNP with the lowest *p*-value was taken as the lead SNP to reduce redundant association signals between different traits and identify candidate genes.

### 2.3. Identification of Significant Loci for Related Traits through GWAS

In this study, the BLUP [38] method was used to analyze the data of six traits to reduce the environmental impact and simplify the calculation. GWAS of grain shape and grain weight (GC, GL, GLWR, GS, GW, and TGW) was carried out using the general linear model (GLM) [39,40]. A stringent criterion of −log10(*p*) > 4.8 was used to determine the association significance of the grain shape and grain weight traits (Appendix A). As a result, 168 QTLs significantly associated with the six traits were found under DS, and 302 QTLs were found under NW, from which a total of 1350 candidate genes were obtained (Appendix A). Among different traits, GS corresponded to the most QTLs (129) while GW had the fewest QTLs (32). In this paper, the QTLs present in both DS and NW environments were selected for the screening of candidate genes. Finally, 16 significant QTLs were selected for further analysis. GLM detected that one, three, two, five, four, and one QTL were significantly associated with GC, GL, GLWR, GS, GW, and TGW, respectively (Table 2). Overall, one QTL region on chromosome 3 (*qGC3.1*) was significantly associated with GC, accounting for 15.62% and 28.75% of the phenotypic variation under DS and NW, respectively (Table 2, Figure 3a,b). Three QTLs were associated with GL, including *qGL3.1* on chromosome 3, *qGL5.1* on chromosome 5, and *qGL8.1* on chromosome 8, which accounted for 19.34% and 14.82%, 15.80% and 18.56%, 14.83% and 15.99% of the phenotypic variation under DS and NW, respectively (Table 2, Figure 3c,d). Two (*qGLWR3.1* and *qGLWR3.2*) of the chromosome 3 QTLs were significantly associated with GLWR, which accounted for 16.32% and 13.28%, 16.63% and 12.98% of the phenotypic variation under DS and NW, respectively (Table 2, Figure 3e,f). Five QTLs were significantly associated with GS, including *qGS2.1* on chromosome 2, *qGS3.1* on chromosome 3, *qGS4.1* and *qGS4.2* on chromosome 4, and *qGS12.1* on chromosome 12, which accounted for 17.72%, 18.82%, 18.81%, 18.77%, and 25.40% of the phenotypic variation under DS, and 24.46%, 23.45%, 23.37%, 23.04%, and 23.37% of the phenotypic variation under NW, respectively. (Table 2, Figure 3g,h). Four QTLs were significantly associated with GW, including *qGW1.1* on chromosome 1, *qGW7.1* on chromosome 7, *qGW8.1* on chromosome 8, and *qGW11.1* on chromosome 11, explaining 16.30–18.35% of phenotypic variation in DS and 15.51%–18.73% of phenotypic variation in NW (Table 2, Figure 3i,j). One QTL on chromosome 2 (*qTGW2.1*) was significantly associated with TGW, accounting for 20.04% and 18.13% of the phenotypic variation in DS and NW, respectively (Table 2, Figure 3k,l).

### 2.4. Candidate Gene Identification and Haplotype Analysis

Sixteen identified QTLs were used for high-density association and gene-based haplotype analysis to identify candidate genes after the removal of genes encoding hypothetical proteins, retrotransposons, and transposon proteins. In the *qGC3.1* region (16.6–16.8 Mb on chromosome 3), 254 SNPs for nine genes were used for high-density association analysis. The gene with the most significant annotation was LOC_Os03g29260 (Figure 4b–d). Based on the two SNPs in the LOC_Os03g29260 promoter, three SNPs in exons, and one SNP in introns, four major haplotypes were detected in 275 *Xian* rice accessions, including ACCCCA for HapA, ACCGCA for HapB, ACTGCA for HapC, and ATTCTA for HapD. The mean GC of HapA, HapB, HapC, and HapD was 18.11, 18.42, 18.66, and 18.00 (mm) in the DS environment and 19.52, 19.71, 20.11, and 19.13 (mm) in the NW environment, respectively (Figure 4a). Haplotype analysis of the whole population showed that the GC of HapC was significantly higher than that of the other three haplotypes in both environments, and the GC in the DS environment was significantly lower than that in the NW environment (Figure 4e,f).

The *qGL5.1* QTL was identified in a 0.8–1.0 Mb region on chromosome 5, and 448 SNPs for 24 genes were used for high-density association analysis. The most significant hit was located at LOC_Os05g02500 (Figure 5b–d). Based on the two SNPs in the exon and one SNP in the intron, three major haplotypes were detected in 275 Xian rice accessions. HapA was AGC, HapB was GAC, and HapC was GGC. The mean GL of HapA, HapB, and HapC was 8.00, 7.84, and 8.08 (mm) in the DS environment and 8.39, 8.23, and 8.51 (mm) in the NW environment, respectively (Figure 5a). Haplotype analysis of the whole population showed that the GL of HapC was significantly higher than that of the other two haplotypes in both environments (Figure 5e,f).

The qGLWR3.2 QTL was detected in a region of 20.9–21.1 Mb on chromosome 3, and 222 SNPs for 12 genes were used for high-density association analysis. The gene with the most significant annotation was LOC_Os03g37930 (Figure 6b–d). Based on one SNP in the exon and six SNPs in the intron of LOC_Os03g37930, two major haplotypes were detected in 275 Xian rice accessions. HapA was ATCCGGC, and HapB was GTCCGAC. The mean GLWR of HapA and hapB was 3.24 and 2.84 in the DS environment and 3.20 and 2.76 in the NW environment, respectively (Figure 6a). Haplotype analysis of the whole population showed that the GLWR of HapA was significantly higher than that of the other haplotype in both environments and that the GLWR in the DS environment was significantly higher than that in the NW environment (Figure 6e,f).

The *qGS4.1* QTL was identified in a 4.3–4.5 Mb region on chromosome 4, and 448 SNPs for 24 genes were used for high-density association analysis. The most significant hit was located in LOC_Os04g08350 (Figure 7b–d). Based on the three SNPs in exons and 20 SNPs in introns of LOC_Os04g08350, seven major haplotypes were detected in 275 Xian rice accessions, including HapA (CAAACCCGTCCTACCGAAAAAGA), HapB (CAAACCCGTCCTACCGTAGGAGA), HapC (TAAACTCGTCCTACCGAAAGGGA), HapD (TAAACTCGTCCTACCGTAAGGGA), HapE (TCAACCCCCGCCCTACCGAAGGGGA), HapF (TCAACCCGTCCTATCGAAAGATG), and HapG (TCATCCCGCCCTACCGTAGGGGA). The mean GS of HapA, HapB, HapC, HapD, HapE, HapF, and HapG was 14.65, 14.63, 14.99, 14.92, 15.90, 15.59, and 15.30 (mm^2^) in the DS environment, and 16.28, 16.81, 16.93, 17.22, 18.12, 17.98, and 17.17 (mm^2^) in the NW environment, respectively. Haplotype analysis of the whole population revealed that the GS of HapE was significantly higher than that of the other six haplotypes in both environments, and HapA, HapB, and HapC were significantly different from HapE and HapF (Figure 7a). Only the DS environment resulted in a significant difference between HapA and HapD (Figure 7e,f).

The *qGW1.1* QTL was detected in a region from 3.4 Mb to 3.6 Mb on chromosome 1, and 292 SNPs for 20 genes were used for high-density association analysis. The gene with the most significant annotation was LOC_Os01g07500 (Figure 8b–d). Three major haplotypes were detected in 275 Xian rice accessions based on the two SNPs in the LOC_Os01g07500 promoter, one SNP in the exon, and 10 SNPs in the intron. HapA was CCCGTGCTGCCAT, HapB was CTCATGCCGCTAT, and HapC was CTCATGTTGCTAT. The mean GW of HapA, HapB, and HapC was 2.74, 2.55, and 2.50 (mm) in the DS environment and 2.96, 2.71, and 2.65 (mm) in the NW environment, respectively (Figure 8a). Haplotype analysis of the whole population showed that the GW of HapA was significantly higher than that of other haplotypes in both environments, and the GW in the DS environment was significantly higher than that in the NW environment. HapA was significantly different from HapB and HapC (Figure 8e,f).

The *qTGW2.1* QTL was identified in an 8.6–8.8 Mb region on chromosome 2, and 306 SNPs for 12 genes were used for high-density association analysis. The most significant hit was located in LOC_Os02g15580 (Figure 9b–d). Based on the three SNPs in exons and the 20 SNPs in introns, two major haplotypes were detected in 275 Xian rice accessions. HapA was A, and HapB was G. The mean GS of HapA and HapB was 16.14 and 16.96 (g) in the DS environment and 22.56 and 24.04 (g) in the NW environment (Figure 9a). Haplotype analysis of the whole population demonstrated that the TGW of HapA was significantly higher than that of HapB in both environments (Figure 9e,f).

## 3. Discussion

Increasing rice yields can help address the problem of global food shortages. Grain shape largely determines grain weight and therefore affects yield and quality [41,42]. GWAS is a genome-wide systematic tool for studying associations between population traits and SNPs based on LD and has been widely used to identify QTLs and genes associated with important traits in many crops [43,44,45].

To the best of our knowledge, this is the first study of GWAS on the grain shape and grain weight of rice under normal and drought-stress conditions. Analysis of phenotypic data identified phenotypic diversity and QTLs associated with rice grain size and grain weight and identified the haplotypes with significant differences in these two indices. In particular, DS resulted in lower values of all six traits except for GLWR than NW, suggesting that DS has obvious negative effects on the grain shape and grain weight of rice. GC (19.90 mm), GL (8.36 mm), GS (17.34 mm^2^), and TGW (23.99 g) were the highest under NW in 2017. These significant phenotypic variants may be associated with high genetic diversity.

Mature seeds are comprised of three primary components: the embryo, endosperm, and seed coat. These components originate from the fertilized egg, the fertilized polar nucleus, and maternal tissue. Rice grains grow inside the spikelet hull with limited caryopsis space. Therefore, grain shape and size are critically determined by the maternal genotype that controls the cell number and size of glumes [21,22]. In addition to the aforementioned pathways that regulate granule shape, cytochrome P450 also plays a crucial role in complex biosynthetic pathways. Specifically, *GE/BG2/GL3.2* encodes a CYP78A family of cytochrome monooxygenases that are primarily expressed in the peltate region, which is located at the boundary between the embryo and endosperm. These enzymes coordinate the growth and development of both the embryo and endosperm, highlighting their importance in these processes. The loss of *GE/BG2/GL3.2* function resulted in increased embryo size and decreased endosperm size, whereas an excess of *GE/BG2/GL3.2* led to decreased embryo size and increased endosperm size [46,47]. The length of grains can be increased by down-regulating the expression of the cytochrome gene CYP704A3. This down-regulation can be achieved through the binding of miRNA to the 3′UTR region of CYP704A3, which regulates its expression and subsequently affects the size of grains [48]. *SRS3/SAR1/OsKINESIN-13A* is a gene that drives vascular deaggregation and is mainly produced by the Golgi apparatus and further distributed to the cell surface via vesicles. The OsKinesin-13A functional mutation disrupted the rotation direction of cellulose microfibers and microtubules, resulting in shorter vascular cells and reduced grain length [49,50]. The protein encoded by UWA contains the NHL structural domain. An increase in the number of fuwa mutant cells leads to wider, thicker, and shorter seeds. The up-regulation of some cell cycle-related genes in fuwa mutants indicates that FUWA is involved in the cell cycle pathway that regulates cell size, ultimately affecting grain size [51]. Here, we identified multiple QTLs for rice grain size and grain weight under normal and drought stress conditions by GWAS using 404,411 high-confidence SNPs. We also found many genes that have been cloned previously, such as *GS3*, *OsGSK3*, and *brd1*, which regulate the GL; *OsRA2*, *GW5,* and *OsDER1*, which control the GW; *LARGE1;* and *OML4*, which regulate the TGW, suggesting that the results of this study are highly reliable. Candidate genes were further identified by haplotype block structure analysis, which showed significant associations with the tested agronomic traits.

For grain shape, five promising candidate genes (one known and four novel) were identified using the 275 *Xian* rice accessions. The first was LOC_Os03g29260 in *qGC3.1*, which was annotated as an elongation factor protein playing an important role in regulating plant growth and development. The second was LOC_Os05g02500 in *qGL5.1*, which was annotated as a dual-specificity protein. This gene has been identified in previous studies as *GSN1* [52], and the *GSN1* mutant showed significant increases in GL, GW, and TGW but a significantly lower grain number per spike and set rate than the wild type, thereby exhibiting a reduction in yield per plant. The third was LOC_Os03g37930 in *qGLWR3.2*. LOC_Os03g37930 was annotated as a potassium transporter, which is thought to have K+ transporter activity, increase abscisic acid (ABA) biosynthesis, and activate the ABA signaling response when encoded. GS5 can also interact with ABA signaling factors [23], thereby affecting rice grain shape. The fourth is LOC_Os04g08350 in *qGS4.1*, which is annotated as a cysteine synthase or chloroplast/chromoplast precursor. Most of the chloroplast proteome is shuttled through the cytoplasm as precursor proteins. In the cytoplasm, the precursors can be co-modified, post-translationally modified, and interact with E3 ligases, while the ubiquitin-proteasome pathway is an important conduit for controlling grain shape. The fifth was LOC_Os01g07500 in *qGW1.1*, which encodes a tryptophan aminotransferase and plays a key role in rice auxin synthesis [53]. For the TGW, we identified a promising gene, LOC_Os02g15580, in *qTGW2.1*. LOC_Os02g15580 is annotated as cyclic nucleotide-gated ion channel 1 and is highly responsive to a variety of stimuli, including hormones such as abscisic acid, indoleacetic acid, agonists, and ethylene, which have a major impact on grain formation and can directly or indirectly control the grain weight.

In summary, this study showed that the GWAS of rice grain shape and grain weight is reliable in different environments. GWAS contributed to the identification of a number of QTLs highly associated with rice grain shape and grain weight in 275 rice accessions, and further haplotype analysis screened the candidate genes related to these agronomic traits. The findings can help understand the genetic basis for rice grain shape and grain weight and also facilitate grain quality improvement in rice breeding programs.

## 4. Materials and Methods

### 4.1. Plant Materials, Field Trials and Trait Measurements

A total of 275 *Xian* rice accessions were obtained from the 3000 Rice Genome Project (3KRGP) [54] (Appendix A).

The 275 accessions were grown at the experiment station of the Institute of Crop Science, Chinese Academy of Agricultural Sciences, Hainan Sanya Nanbin Farm (18.3° N, 109.3° E) from December 2016 to May 2017 and from December 2017 to May 2018, respectively. Two treatments of normal water (NW) and drought stress (DS) were carried out, with each treatment comprising two replications. Each material was planted in two rows with 10 plants in each row and a planting spacing of 20 cm × 25 cm.

For DS treatment, when the transplants were 25-days-old, no water except for a small amount of natural rainfall was supplied until maturity. For NW treatment, the fields were treated as general field management. Field management practices were consistent with local management standards. When the rice is fully mature, the seeds are harvested and dried in the sun. The grain circumference (GC) (mm), grain length (GL) (mm), grain width (GW) (mm), grain length to width ratio (GLWR), and grain size (GS) (mm^2^) were measured using a rice grain appearance quality scanning machine (Model SC-G, Hangzhou, China), and the thousand grain weight (TGW) (g) was determined with a high-precision electronic balance (1/1000, APTP456 series). Before each measurement, the instrument was calibrated with a calibration target.

### 4.2. Statistical Analysis

Data were collated using Excel 2018 and SPSS, and means, standard deviations, and coefficients of variation were calculated for each trait. Correlation and frequency analyses were conducted for traits related to rice grain shape and grain weight, and the best linear unbiased prediction (BLUP) values for six traits of rice grain shape and grain weight over two years were used for GWAS.

### 4.3. Genetic Fractal and Population Structure Analysis

Genotype data were obtained from high-density SNP data from the 3K RGP database rice SNP-seek database (http://snp-seek.irri.org/, accessed on 1 March 2023) [6,55,56], and SNPs were further screened using PLINK (version 1.9) [57,58]. MAF > 0.05, GENO < 0.2, for a total of 404,411 SNPs.

The TASSEL [59] was used to calculate the population structure (Q) and kinship (K). Based on the SNPs of 275 rice accessions, we calculated the genetic distance matrix using VCF2Dis (https://github.com/BGI-shenzhen/VCF2Dis, accessed on 6 March 2023). Individual-based neighbor joining (NJ) trees were constructed and embellished using iTOL (https://itol.embl.de/, accessed on 7 March 2023). Principal component analysis (PCA) was performed using GCTA (version 1.93.2) software [60] on Linux, and principal components were plotted using the R package ‘ggplot2’. The PCA scores and relationship matrix will be used in the general linear model (GLM) below [39]. To estimate LD in the rice population, the value of the squared correlation coefficient (r^2^) between pairs of SNPs was calculated using PopLDdecay decay [61], and the distance across the chromosome when the r^2^ dropped to half of its maximum value was called the LD decay distance [19].

### 4.4. Genome-Wide Association Mapping

In this study, we obtained 404,411 SNPs (MAF > 0.05) and six sets of phenotypic data to perform GWAS in TASSEL (version 5.2.40) software using the GLM. For the GLM, the effective independent SNP number calculated using GEC (Version 1.0) [62] software was used to determine the threshold. *p*-value = 1.4 × 10^−5^ was chosen as the significant threshold, and points with *p*-value less than the threshold were considered SNPs associated with the phenotype, and adjacent significant SNPs associated with the same trait within a physical distance of 200 kb were considered one candidate region. The Manhattan plot was drawn using the R package “CMplot”.

### 4.5. Identification of Candidate Genes and Haplotype Analysis

To identify the candidate genes associated with GC, GL, GLWR, GS, GW, and TGW, the Rice Genome Annotation Project (http://rice.plantbiology.msu.edu, accessed on 26 March 2023) was used to search for candidate genes in the 200 kb genomic region of the selected SNPs. Among all the candidate genes, four types of genes were excluded: expressed proteins, hypothetical proteins, retrotransposons, and transposons. Further haplotype analysis excluded heterozygotes and missing alleles, and Haplogroups consisting of fewer than 10 accessions were deleted. For the genes found in QTLs, only non-synonymous SNPs in the coding regions of these genes were used for haplotype analysis of R, and a Student’s *t*-test was performed to determine whether this locus could cause changes in rice grain shape and grain weight. The variable r was used to visualize the results.

## 5. Conclusions

This study determined the grain shape and grain weight traits (GC, GL, GLWR, GS, GW, and TGW) of 275 rice indica accessions in two environments and two years. By combining GWAS with the genotype data, a total of 16 important QTLs were screened out. Haplotype differential analysis and functional annotation of the candidate genes among these QTLs identified one cloned gene (*GSN1*) and five candidate genes (LOC_Os03g29260, LOC_Os03g37930, LOC_Os04g08350, LOC_Os01g07500, and LOC_Os02g15580). The findings can enrich the gene resource for rice grain shape and weight and provide valuable references for future molecular breeding of rice.

## Figures and Tables

**Figure 1 plants-12-02549-f001:**
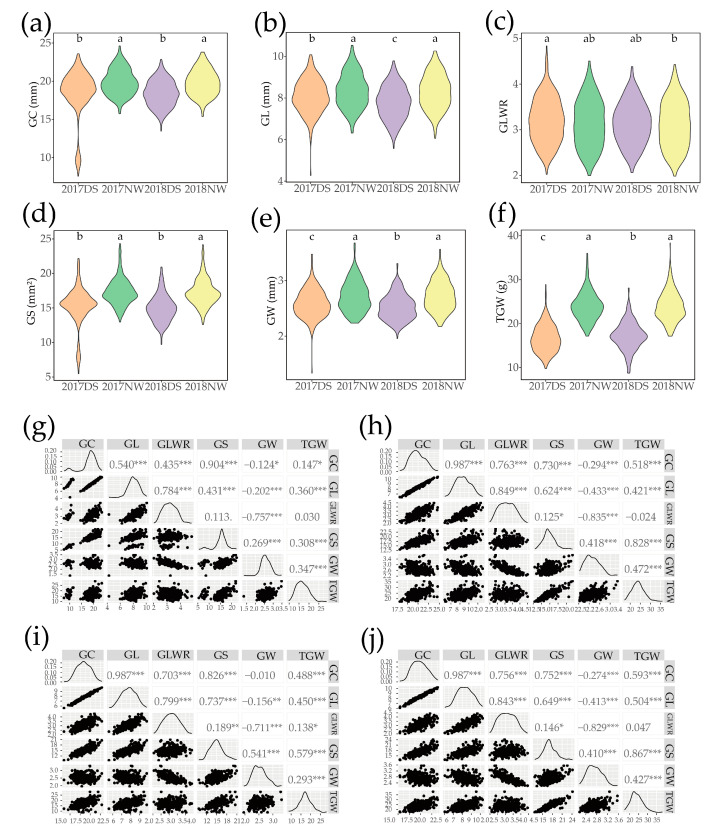
Box plots of six traits of rice grain shape and weight under drought stress (DS) and normal water (NW) and phenotypic correlations of the six traits in different environments. 2017DS: 2017 drought stress; 2017NW: 2017 normal water; 2018DS: 2018 drought stress; 2018NW: 2018 normal water. (**a**) Grain circumference; (**b**) grain length; (**c**) grain length to width ratio; (**d**) grain size; (**e**) grain width; (**f**) thousand grain weight; (**g**) 2017 drought stress; (**h**) 2017 normal water; (**i**) 2018 drought stress; (**j**) 2018 normal water. ‘a’, ‘b’ and ‘c’ are based on whether the *t*-test is significant between each other. ‘*’, ‘**’, and ‘***’ refer to significant correlations (*p* < 0.05, *p* < 0.01, and *p* < 0.001).

**Figure 2 plants-12-02549-f002:**
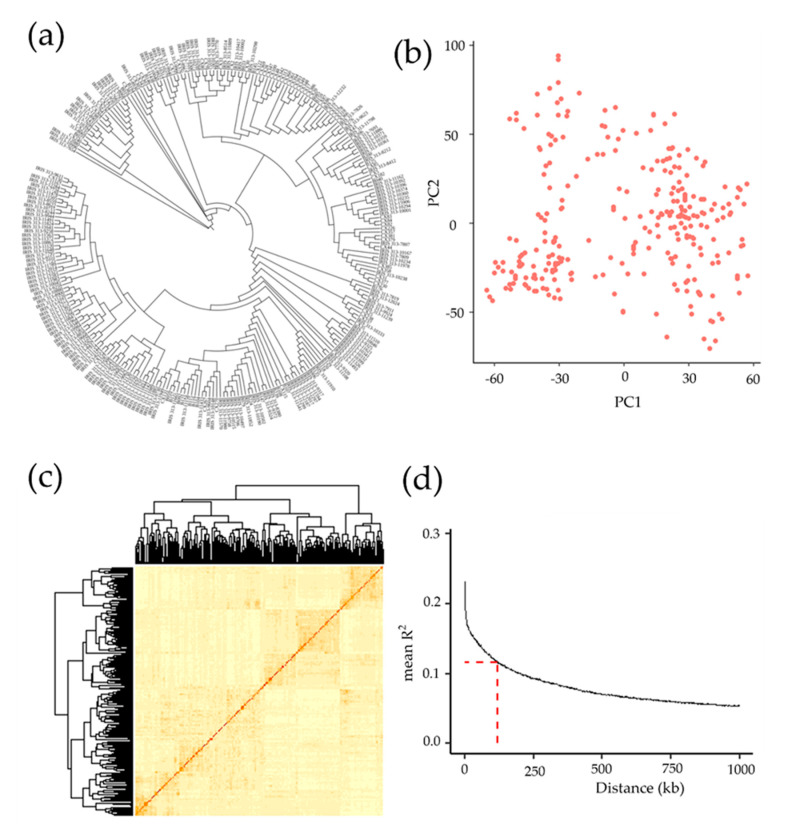
Genetic structure analysis of 275 *Xian* rice accessions. (**a**) Phylogenetic tree; each branch corresponds to a rice accession. (**b**) Principal component analysis on about 0.4 million SNPs in 275 rice accessions. PC1 and PC2 refer to the first and second principal components, respectively. Red points represent the 275 rice accessions, with each point representing one rice accession. A shorter distance between the points indicates a closer relationship. (**c**) Heatmap of kinship from R Package “pheatmap”. (**d**) LD decay. Y–axis is the average r^2^ value of each 250−kb region, and X–axis is the physical distance between markers.

**Figure 3 plants-12-02549-f003:**
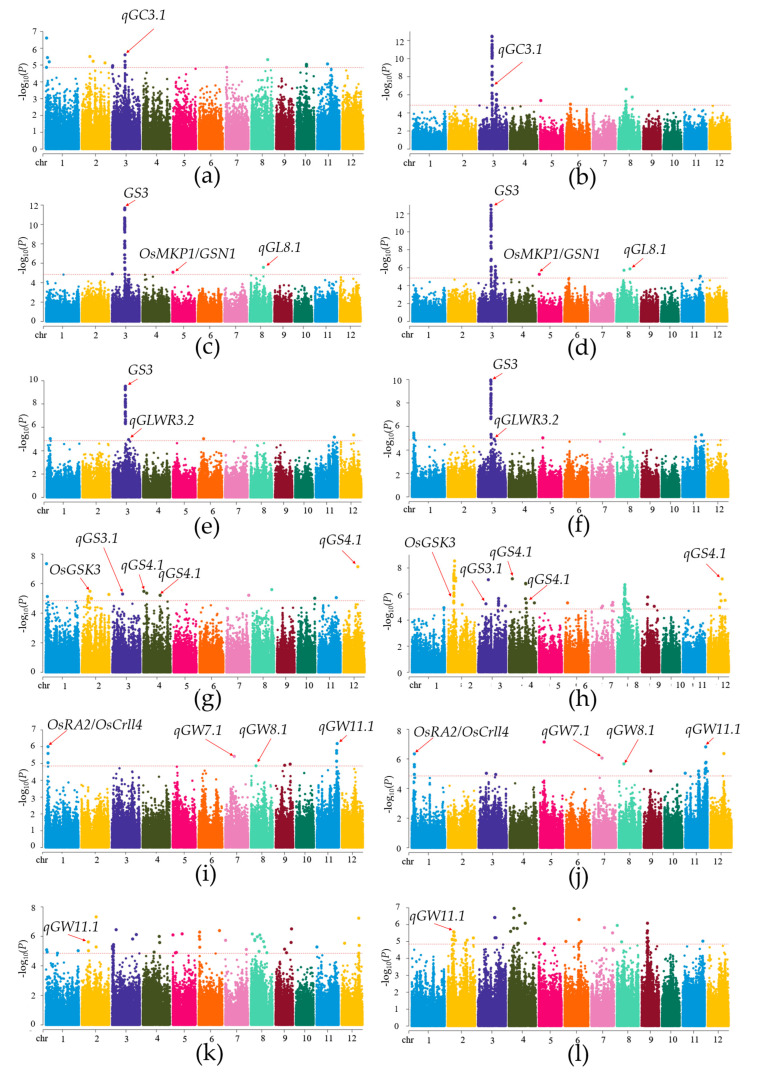
Manhattan plots of the GWAS results for GC, GL, GLWR, GS, GW, and TGW obtained with GLM. (**a**) GC under drought stress; (**b**) GC under normal water; (**c**) GL under drought stress; (**d**) GL under normal water; (**e**) GLWR under drought stress; (**f**) GLWR under normal water; (**g**) GS under drought stress; (**h**) GS under normal water; (**i**) GW under drought stress; (**j**) GW under normal water; (**k**) TGW under drought stress; (**l**) TGW under normal water.

**Figure 4 plants-12-02549-f004:**
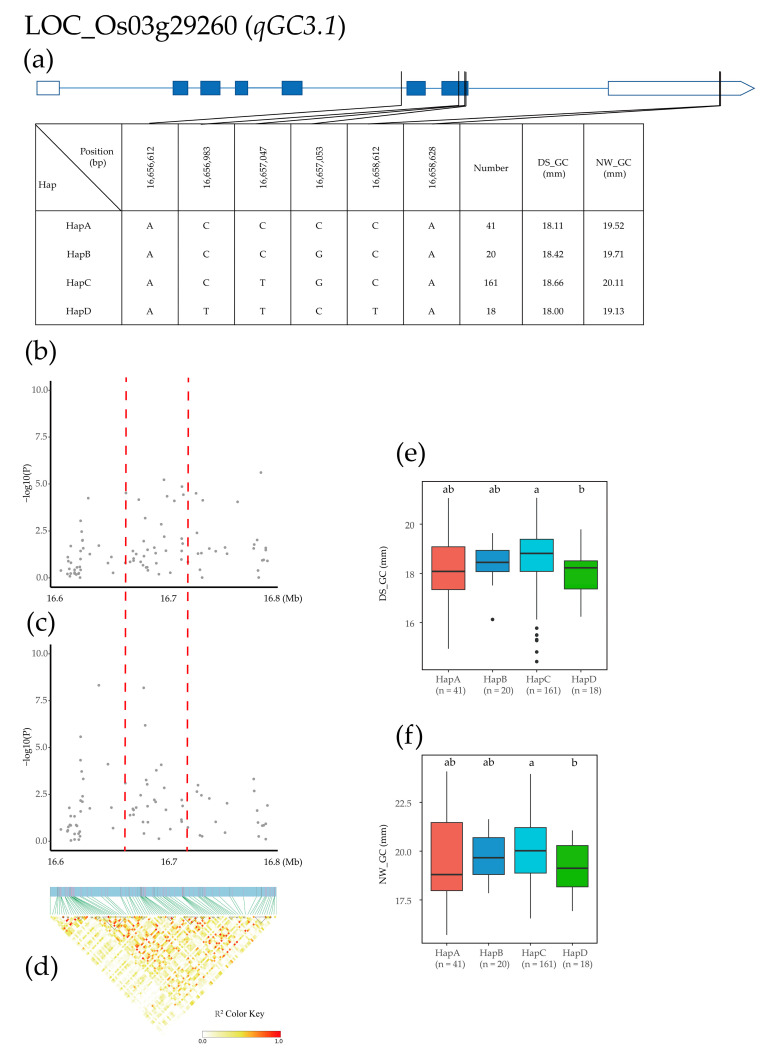
Identification of candidate genes for GC. (**a**) Based on six SNPs in all evaluated rice accessions, four haplotypes of LOC_Os03g29260 were identified. In the gene structure diagram of LOC_Os03g29260, the promoter is indicated by white frame; the exon is represented by blue frame; and the intron and intergenic region are marked by blue lines. A thin black line represents the genomic location of each SNP. Haplotypes with fewer than 10 accessions are not shown. (**b**,**c**) Local Manhattan map under drought stress and normal water. Red dotted lines represent candidate regions for associated SNPs. (**d**) Linkage disequilibrium heatmap. (**e**,**f**) Based on GL of LOC_Os06g15480 haplotype under drought stress and normal water, differences between haplotypes were statistically analyzed using Tukey’s test, ‘a’ and ‘b’ are based on whether the *t*-test is significant between each other.

**Figure 5 plants-12-02549-f005:**
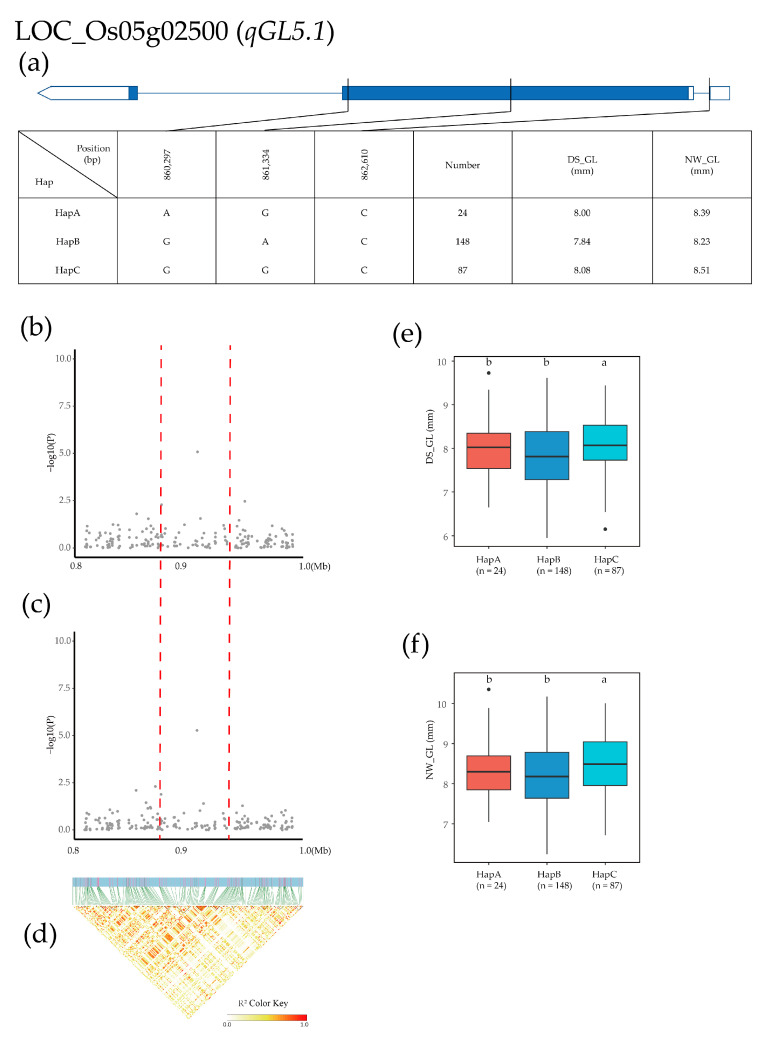
Dentification of candidate genes for GL. (**a**) Based on the three SNPs in all evaluated rice accessions, three haplotypes of LOC_Os05g02500 were identified. In the gene structure diagram of LOC_Os05g02500, the promoter is indicated by white frame; the exon is represented by blue frame; and the intron and intergenic region are marked by blue lines. A thin black line represents the genomic location of each SNP. Haplotypes with fewer than 10 accessions are not shown. (**b**,**c**) Local Manhattan map under drought stress and normal water. Red dotted lines represent candidate regions for associated SNPs. (**d**) Linkage disequilibrium heatmap. (**e**,**f**) Based on GL of LOC_Os05g02500 haplotype under drought stress and normal water, differences between haplotypes were statistically analyzed using Tukey’s test, ‘a’ and ‘b’ are based on whether the *t*-test is significant between each other.

**Figure 6 plants-12-02549-f006:**
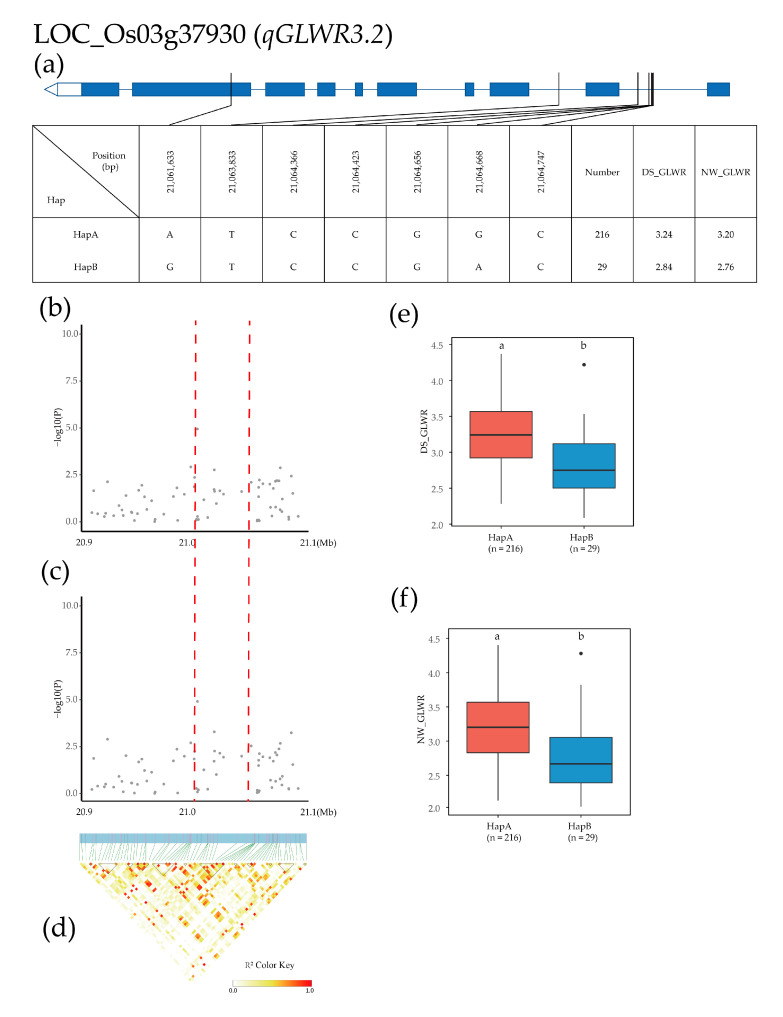
Identification of candidate genes for GLWR. (**a**) Based on the tree SNPs in all evaluated rice accessions, two haplotypes of LOC_Os03g37930 were identified. In the gene structure diagram of LOC_Os03g37930, the promoter is indicated by white frame; the exon is represented by blue frame; and the intron and intergenic region are marked by blue lines. A thin black line represents the genomic location of each SNP. Haplotypes with fewer than 10 accessions are not shown. (**b**,**c**) Local Manhattan map under drought stress and normal water. Red dotted lines represent candidate regions for associated SNPs. (**d**) Linkage disequilibrium heatmap. (**e**,**f**) Based on GL of LOC_Os03g37930 haplotype under drought stress and normal water, differences between haplotypes were statistically analyzed using Tukey’s test, ‘a’ and ‘b’ are based on whether the *t*-test is significant between each other.

**Figure 7 plants-12-02549-f007:**
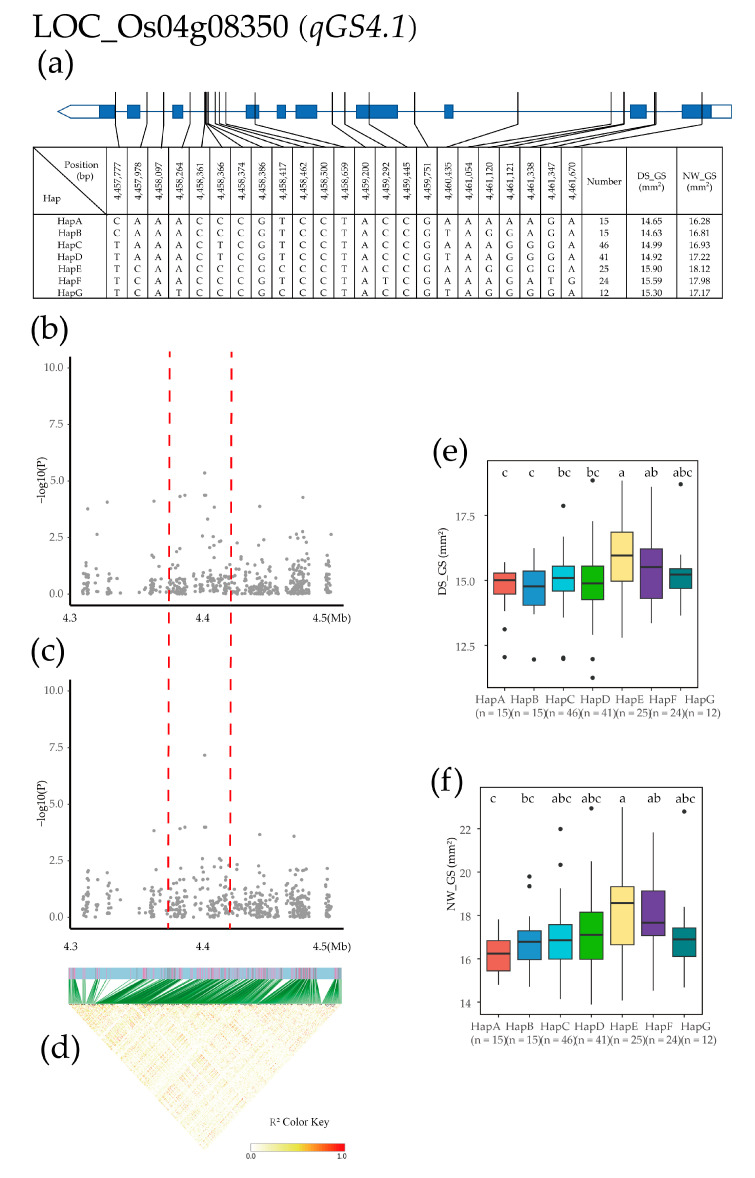
Identification of candidate genes for GS. (**a**) Based on 24 SNPs in all evaluated rice accessions, seven haplotypes of LOC_Os04g08350 were identified. In the gene structure diagram of LOC_Os04g08350, the promoter is indicated by white frame; the exon is represented by blue frame; and the intron and intergenic region are marked by blue lines. A thin black line represents the genomic location of each SNP. Haplotypes with fewer than 10 accessions are not shown. (**b**,**c**) Local Manhattan map under drought stress and normal water. Red dotted lines represent candidate regions for associated SNPs. (**d**) Linkage disequilibrium heatmap. (**e**,**f**) Based on GL of LOC_Os04g08350 haplotype under drought stress and normal water, differences between haplotypes were statistically analyzed using Tukey’s test, ‘a’, ‘b’ and ‘c’ are based on whether the *t*-test is significant between each other.

**Figure 8 plants-12-02549-f008:**
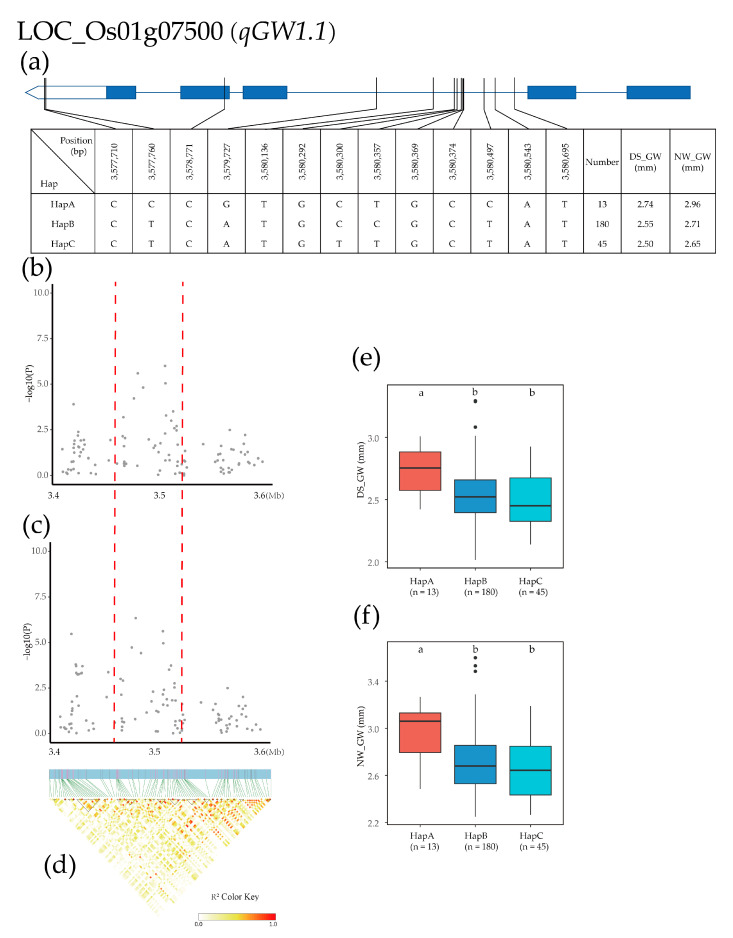
Identification of candidate genes for GW. (**a**) Based on 13 SNPs in all evaluated rice accessions, three haplotypes of LOC_Os01g07500 were identified. In the gene structure diagram of LOC_Os01g07500, the promoter is indicated by white frame; the exon is represented by blue frame; and the intron and intergenic region are marked by blue lines. A thin black line represents the genomic location of each SNP. Haplotypes with fewer than 10 accessions are not shown. (**b**,**c**) Local Manhattan map under drought stress and normal water. Red dotted lines represent candidate regions for associated SNPs. (**d**) Linkage disequilibrium heatmap. (**e**,**f**) Based on GL of LOC_Os04g08350 haplotype under drought stress and normal water, differences between haplotypes were statistically analyzed using Tukey’s test, ‘a’ and ‘b’ are based on whether the *t*-test is significant between each other.

**Figure 9 plants-12-02549-f009:**
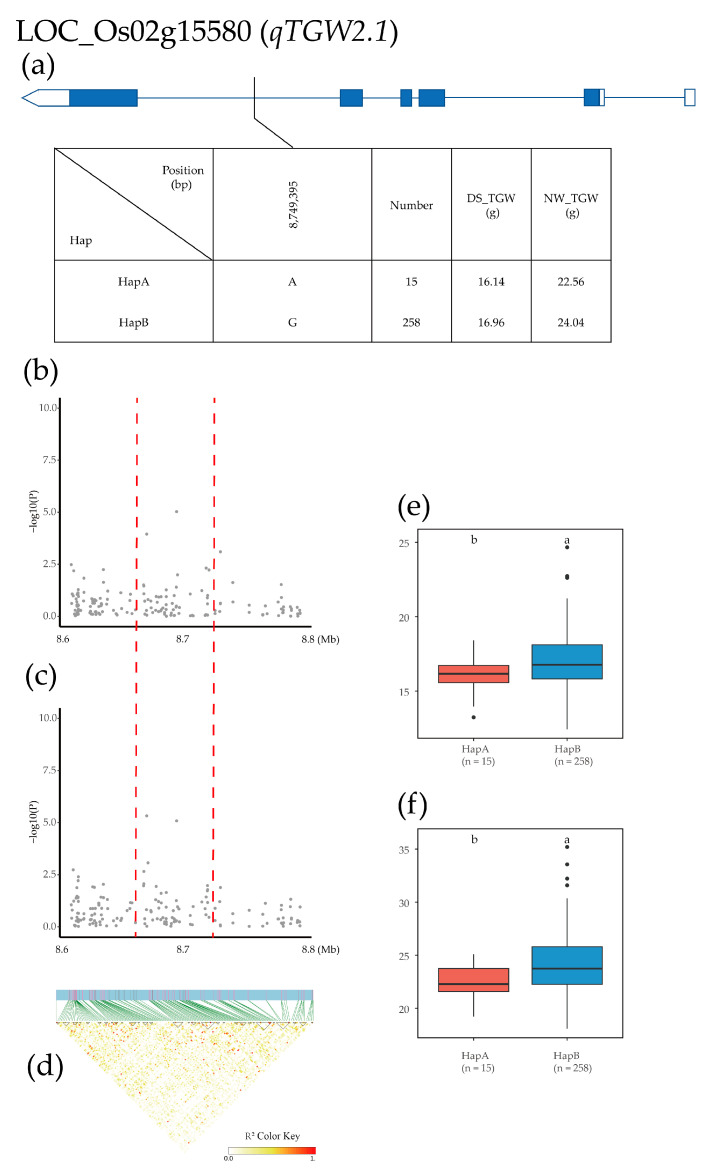
Identification of candidate genes for TGW. (**a**) Based on one SNP in all evaluated rice accessions, two haplotypes of LOC_Os02g15580 were identified. In the gene structure diagram of LOC_Os02g15580, the promoter is indicated by white frame; the exon is represented by blue frame; and the intron and intergenic region are marked by blue lines. A thin black line represents the genomic location of each SNP. Haplotypes with fewer than 10 accessions are not shown. (**b**,**c**) Local Manhattan map under drought stress and normal water. Red dotted lines represent candidate regions for associated SNPs. (**d**) Linkage disequilibrium heatmap. (**e**,**f**) Based on GL of LOC_Os02g15580 haplotype under drought stress and normal water, differences between haplotypes were statistically analyzed using Tukey’s test, ‘a’ and ‘b’ are based on whether the *t*-test is significant between each other.

**Table 1 plants-12-02549-t001:** Statistics of GC, GL, GLWR, GS, GW, and TGW in different environments.

	Treatment	Year	Mean ± SD	Max	Min	CV (%)	H^2^
GC (mm)	DS	2017	18.53 ± 3.05	23.59	7.58	16.44%	0.72
		2018	18.38 ± 1.68	22.86	13.48	9.11%	
	NW	2017	19.90 ± 1.67	24.60	15.74	8.39%	0.97
		2018	19.78 ± 1.66	23.80	15.39	8.42%	
GL (mm)	DS	2017	8.09 ± 0.80	10.08	4.27	9.84%	0.94
		2018	7.78 ± 0.79	9.79	5.57	10.13%	
	NW	2017	8.36 ± 0.83	10.53	6.32	9.91%	0.98
		2018	8.31 ± 0.82	10.27	6.07	9.89%	
GLWR	DS	2017	3.21 ± 0.49	4.84	2.02	15.28%	0.97
		2018	3.14 ± 0.45	4.38	2.06	14.46%	
	NW	2017	3.14 ± 0.52	4.51	1.99	16.69%	0.98
		2018	3.11 ± 0.51	4.43	1.98	16.43%	
GS (mm^2^)	DS	2017	15.26 ± 2.76	22.16	5.46	18.08%	0.68
		2018	14.97 ± 1.92	20.91	9.70	12.81%	
	NW	2017	17.34 ± 1.90	24.31	12.95	10.97%	0.96
		2018	17.25 ± 1.91	24.15	12.6	11.06%	
GW (mm)	DS	2017	2.59 ± 0.25	3.48	1.33	9.71%	0.95
		2018	2.52 ± 0.23	3.31	1.96	8.95%	
	NW	2017	2.72 ± 0.27	3.68	2.23	9.94%	0.98
		2018	2.73 ± 0.26	3.56	2.18	9.66%	
TGW(g)	DS	2017	16.67 ± 3.04	28.88	9.79	18.26%	0.71
		2018	17.21 ± 3.16	28.10	8.75	18.38%	
	NW	2017	23.99 ± 3.26	35.99	17.18	13.58%	0.99
		2018	23.96 ± 3.32	38.33	17.17	13.87%	

Note: GC, grain circumference; GL, grain length; GLWR, grain length to width ratio; GS, grain size; GW, grain width; TGW, thousand grain weight.

**Table 2 plants-12-02549-t002:** Sixteen QTLs of significant associations with GC, GL, GLWR, GS, GW, and TGW.

Trait	QTL	Chr	Lead SNP (bp)	DS R^2^ (%)	NW R^2^ (%)	DS *p*-Value	NW *p*-Value	Known Genes/QTLs
GC	*qGC3.1*	3	16,743,121	15.62%	28.75%	6 × 10^−6^	6.15 × 10^−11^	
GL	*qGL3.1*	3	16,725,044	19.34%	18.56%	1.36 × 10^−7^	1.57 × 10^−7^	*GS3*
	*qGL5.1*	5	955,623	14.82%	14.83%	8.4 × 10^−6^	5.38 × 10^−6^	*OsMKP1*; *GSN1*
	*qGL8.1*	8	18,001,936	15.80%	15.99%	2.63 × 10^−6^	1.35 × 10^−6^	
GLWR	*qGLWR3.1*	3	16,725,044	16.32%	16.63%	4.1 × 10^−7^	2.19 × 10^−7^	*GS3*
	*qGLWR3.2*	3	21,067,616	13.28%	12.98%	1.14 × 10^−5^	1.23 × 10^−5^	
GS	*qGS2.1*	2	7,792,944	17.72%	24.46%	1.09 × 10^−5^	4.70 × 10^−8^	*OsGSK3*
	*qGS3.1*	3	13,224,102	18.82%	23.45%	5.04 × 10^−6^	8.07 × 10^−8^	
	*qGS4.1*	4	4,445,624	18.81%	23.37%	4.40 × 10^−6^	6.75 × 10^−8^	
	*qGS4.2*	4	21,851,870	18.77%	23.04%	6.06 × 10^−6^	1.55 × 10^−7^	
	*qGS12.1*	12	18,698,797	25.40%	23.37%	7.17 × 10^−8^	7.01 × 10^−8^	
GW	*qGW1.1*	1	3,526,032	17.91%	15.51%	9.99 × 10^−7^	2.42 × 10^−6^	*OsRA2*; *OsCrll4*
	*qGW7.1*	7	11,595,349	16.30%	16.49%	3.71 × 10^−6^	8.76 × 10^−7^	
	*qGW8.1*	8	6,558,586	18.35%	18.73%	1.38 × 10^−5^	2.17 × 10^−6^	
	*qGW11.1*	11	26,572,137	18.20%	18.18%	6.51 × 10^−7^	1.55 × 10^−7^	
TGW	*qTGW2.1*	2	8,783,659	20.04%	18.13%	9.35 × 10^−6^	8.08 × 10^−6^	

## Data Availability

Not applicable.

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
