# Peer review of "Genome-Wide Association Study of Xian Rice Grain Shape and Weight in Different Environments"

_plants, 2023, doi:10.3390/plants12132549_

Round 1
Reviewer 1 Report
The manuscript entitled,"Genome-wide association study of xian rice grain shape and weight in different environments" by Nansheng et al presents a GWAS case in which candidate genes for grain size and shape related traits have been identified.
Authors have obtained 275 xian rice accessions alongwith their genotype data from the 3K rice genome project. I could n't find the structured sub-populations within germplasm set of 275 accessions. I have only one objective question, on what basis the set of 275 accessions were selected for the study ? On what hypothesis the study was design ?
Author Response
Manuscript report
The manuscript entitled,"Genome-wide association study of xian rice grain shape and weight in different environments" by Nansheng et al presents a GWAS case in which candidate genes for grain size and shape related traits have been identified.
Response: Thank you for the positive comments.
- The Authors have obtained 275 xian rice accessions along with their genotype data from the 3K rice genome project. I couldn't find the structured sub-populations within germplasm set of 275 accessions. I have only one objective question, on what basis the set of 275 accessions were selected for the study? On what hypothesis the study was design?
The manuscript entitled,"Genome-wide association study of xian rice grain shape and weight in different environments" by Nansheng et al presents a GWAS case in which candidate genes for grain size and shape related traits have been identified.
Response: Thank you for the positive comments.
Authors Response
Point-by-point responses to the reviewers’ comments:
- The Authors have obtained 275 xian rice accessions along with their genotype data from the 3K rice genome project. I couldn't find the structured sub-populations within germplasm set of 275 accessions. I have only one objective question, on what basis the set of 275 accessions were selected for the study? On what hypothesis the study was design?
回应:感谢您的评论。该研究使用了来自275K水稻基因组重测序计划的3种水稻材料,如图2(a)和(b)所示。275种材料产地不同,遗传变异广泛,基因型代表性较好;从图S1可以看出,这组材料的相关表型变异较大,代表性也较好。从基因型和表型的角度来看,这套材料适用于通过GWAS分析进行造粒相关基因探索。由于项目组拥有基因型和表型变异类型丰富、来源广泛的材料,项目组正在开展水稻籽粒形状改良和品质育种,通过多环境表型鉴定并结合基因组重测序结果,开展水稻籽粒形状和籽粒重相关基因挖掘,为稻粒型和籽粒重定向改良奠定了基础。

Reviewer 2 Report
Comments:
1. The names of QTL and genes should be in italics, please check. Please apply superscript for specific text, such as r2 in line 145, mm2 in line 284 and ....
2. Please re-calculate the R2 of each QTL in Table 2, especially for GS. How the total phenotypic variation of the 5 QTL (qGS2.1, qGS3.1, qGS4.1, qGS4.2, qGS12.1) under NW was more than 100%?
3. Please check all the significant levels in Fig.4 to Fig.9, most of them were not reasonable. For instance, the significant levels in Fig.4 (e)(f) were not matched with the result in Fig.4 (a). Also, what does significant level "ac" mean in Fig.4 (e)?
No comments.
Author Response
Manuscript report
- The names of QTL and genes should be in italics, please check. Please apply superscript for specific text, such as r2 in line 145, mm2 in line 284 and ....
- Please re-calculate the R2 of each QTL in Table 2, especially for GS. How the total phenotypic variation of the 5 QTL (1, qGS3.1, qGS4.1, qGS4.2, qGS12.1) under NW was more than 100%?
- Please check all the significant levels in Fig.4 to Fig.9, most of them were not reasonable. For instance, the significant levels in Fig.4 (e)(f) were not matched with the result in Fig.4 (a). Also, what does significant level "ac" mean in Fig.4 (e)?
Authors Response
Point-by-point responses to the reviewers’ comments:
- The names of QTL and genes should be in italics, please check. Please apply superscript for specific text, such as r2 in line 145, mm2 in line 284 and ....
Response: Thank you for your comments. This issue has been modified in the text.
- Please re-calculate the R2 of each QTL in Table 2, especially for GS. How the total phenotypic variation of the 5 QTL (qGS2.1, qGS3.1, qGS4.1, qGS4.2, qGS12.1) under NW was more than 100%?
Response: Thank you for your comments. The association studies in TASSEL (GLM) are performed on markers one at a time. Therefore, the sum of the R square (R2) of markers could be bigger than 100%. One of the reasons is due to linkage disequilibrium (LD) between markers. For example, if a marker has R2 of 20% and the marker is in complete LD with other five markers, then the five markers will have R2 sum to 120%.
- Please check all the significant levels in Fig.4 to Fig.9, most of them were not reasonable. For instance, the significant levels in Fig.4 (e)(f) were not matched with the result in Fig.4 (a). Also, what does significant level "ac" mean in Fig.4 (e)?
Response: Thank you for your comments. This issue has been modified in the text.

Reviewer 3 Report
Dear authors, I am very happy to review your manuscript entitled "Genome-wide association study of xian rice grain shape and weight in different environments", in which you re-sequenced 275 xian (I think indica) accessions and then conducted GWAS on six agronomic traits including grain circumference (GC), grain length (GL), grain length to width ratio (GLWR), grain size (GS), grain width (GW), and thousand grain weight (TGW) with the 404,411 derived SNPs.
In current study and analysis of two-year data, authors have identified 16 QTLs associated with rice grain shape and grain weight that were positioned on chromosomes 1, 2, 3, 4, 5, 7, 8, 11 and 12. Howerver, SNPs-based further analysis made the authors to finalise one cloned (GSN1) and five new candidate genes involved in rice grain shape and weight under drought stress conditions, suggesting a complete and well-organised study.
I believe the current version of the article is quite fine in all aspects, however I have only two suggestions, which I think should be followed before further processing.
1- Two groups are clearly visible in Fig 2b: one near to Y axis (PC2) between -50 to 0 and second from 0-60 on X axis (PC1). How would you justify it?
2- Please improve the font size of axis of graphs in Figure 3
3- Discussion needs some improvements as it is full of results and lack some evidences. Hence, improve it.
4- No ref from 2023 is included that shows the literature review may have some gaps. I would suggest to revise the literature carefully and add some ref from 2023 and 2022.
Best wishes,
Author Response
Manuscript report
Dear authors, I am very happy to review your manuscript entitled "Genome-wide association study of xian rice grain shape and weight in different environments", in which you re-sequenced 275 xian (I think indica) accessions and then conducted GWAS on six agronomic traits including grain circumference (GC), grain length (GL), grain length to width ratio (GLWR), grain size (GS), grain width (GW), and thousand grain weight (TGW) with the 404,411 derived SNPs.
In current study and analysis of two-year data, authors have identified 16 QTLs associated with rice grain shape and grain weight that were positioned on chromosomes 1, 2, 3, 4, 5, 7, 8, 11 and 12. Howerver, SNPs-based further analysis made the authors to finalise one cloned (GSN1) and five new candidate genes involved in rice grain shape and weight under drought stress conditions, suggesting a complete and well-organised study.
I believe the current version of the article is quite fine in all aspects, however I have only two suggestions, which I think should be followed before further processing.
- Two groups are clearly visible in Fig 2b: one near to Y axis (PC2) between -50 to 0 and second from 0-60 on X axis (PC1). How would you justify it?
- Please improve the font size of axis of graphs in Figure 3?
- Discussion needs some improvements as it is full of results and lack some evidences. Hence, improve it.
- No ref from 2023 is included that shows the literature review may have some gaps. I would suggest to revise the literature carefully and add some ref from 2023 and 2022.
Dear authors, I am very happy to review your manuscript entitled "Genome-wide association study of xian rice grain shape and weight in different environments", in which you re-sequenced 275 xian (I think indica) accessions and then conducted GWAS on six agronomic traits including grain circumference (GC), grain length (GL), grain length to width ratio (GLWR), grain size (GS), grain width (GW), and thousand grain weight (TGW) with the 404,411 derived SNPs.
In current study and analysis of two-year data, authors have identified 16 QTLs associated with rice grain shape and grain weight that were positioned on chromosomes 1, 2, 3, 4, 5, 7, 8, 11 and 12. Howerver, SNPs-based further analysis made the authors to finalise one cloned (GSN1) and five new candidate genes involved in rice grain shape and weight under drought stress conditions, suggesting a complete and well-organised study.
I believe the current version of the article is quite fine in all aspects, however I have only two suggestions, which I think should be followed before further processing.
Response: Thank you for the positive comments.
Authors Response
Point-by-point responses to the reviewers’ comments:
- Two groups are clearly visible in Fig 2b: one near to Y axis (PC2) between -50 to 0 and second from 0-60 on X axis (PC1). How would you justify it?
Response: Thank you for your comments. There were four XI clusters (XI-1A from East Asia, XI-1B of modern varieties of diverse origins, XI-2 from South Asia and XI-3 from Southeast Asia)[1]. However, in the analysis of the results, we will only use a large group of xian[2-4].
- Please improve the font size of axis of graphs in Figure 3?
Response: Thank you for your comments. Fig 3 has been modified in the text.
- Discussion needs some improvements as it is full of results and lack some evidences. Hence, improve it.
Response: Thank you for your comments. The discussion has been corrected in the text.
- No ref from 2023 is included that shows the literature review may have some gaps. I would suggest to revise the literature carefully and add some ref from 2023 and 2022.
Response: Thank you for your comments. Some references for 2023 and 2022 have been added to the text. Four in 2023 and six in 2022.
References
- Genomic variation in 3,010 diverse accessions of Asian cultivated rice. Nature 2018.
- Xiang, J.; Zhang, C.P.; Wang, N.S.; Liang, Z.J.; Zhenzhen, Z.; Liang, L.P.; Yuan, H.Y.; Shi, Y.Y. Genome-Wide Association Study Reveals Candidate Genes for Root-Related Traits in Rice. Curr. Issues Mol. Biol. 2022, 44, 4386-4405, doi:10.3390/cimb44100301.
- Wang, Y.; Liu, H.; Meng, Y.; Liu, J.; Ye, G. Validation of genes affecting rice mesocotyl length through candidate association analysis and identification of the superior haplotypes. Frontiers in plant science 2023, 14, 1194119, doi:10.3389/fpls.2023.1194119.
- Pan, Y.H.; Chen, L.; Zhu, X.Y.; Li, J.C.; Rashid, M.A.R.; Chen, C.; Qing, D.J.; Zhou, W.Y.; Yang, X.H.; Gao, L.J.; et al. Utilization of natural alleles for heat adaptability QTLs at the flowering stage in rice. BMC Plant Biol. 2023, 23, 16, doi:10.1186/s12870-023-04260-5.

Round 2
Reviewer 1 Report
Authors response is incomplete. Authors must explain the three genetic groups with respect to phenotypic OR geographic OR any other parameter. These parameters (phenotypic OR geographic OR any other) should coroborate genetic diversity.
Author Response
Manuscript report
- Authors response is incomplete. Authors must explain the three genetic groups with respect to phenotypic OR geographic OR any other parameter. These parameters (phenotypic OR geographic OR any other) should coroborate genetic diversity.
Authors Response
Point-by-point responses to the reviewers’ comments:
- Authors response is incomplete. Authors must explain the three genetic groups with respect to phenotypic OR geographic OR any other parameter. These parameters (phenotypic OR geographic OR any other) should coroborate genetic diversity.
Response: Thank you for your comments. A total of 275 xian rice accessions were obtained from the 3000 Rice Genome Project (3KRGP). As shown in Table S1, these 275 accessions come from different countries and regions.
It can be seen from Fig 2(a) and (b), the set of accessions have extensive genetic variation, and genotypes were well represented.
It can be seen from Fig S1, the phenotype distribution of the group of accessions is relatively wide and representative.
From the perspective of genotype and phenotype, this set of accessions is suitable for grain shape and grain weight gene exploration through GWAS analysis. Since the project team has accessions with abundant genotype and phenotypic variant types and a wide range of sources, and the project team is carrying out rice grain shape improvement and quality breeding, through multi-environment phenotypic identification and combined with genome resequencing results, rice grain shape and grain weight related gene mining is carried out, which lays the foundation for rice grain shape and grain weight improvement.
The content of the last answer was copied wrong. I am very sorry for the trouble. Thank you very much.

Round 3
Reviewer 1 Report
ms can be accepted now.